# Source Control of Gram-Negative Bacteria Using Self-Disinfecting Sinks in a Swedish Burn Centre

**DOI:** 10.3390/microorganisms11040965

**Published:** 2023-04-07

**Authors:** Maria Gideskog, Tina Falkeborn, Jenny Welander, Åsa Melhus

**Affiliations:** 1Department of Communicable Disease and Infection Control, Linköping University Hospital, SE-581 85 Linköping, Sweden; 2Department of Biomedical and Clinical Sciences, Linköping University, SE-581 85 Linköping, Sweden; 3Department of Clinical Microbiology, Linköping University Hospital, SE-581 85 Linköping, Sweden; 4Section of Clinical Microbiology, Department of Medical Sciences, Uppsala University, SE-751 85 Uppsala, Sweden

**Keywords:** sink, water trap, bacterial transmission, self-disinfecting sink, infection control, *Pseudomonas aeruginosa*, *Stenotrophomonas maltophilia*, *Acinetobacter*

## Abstract

Several retrospective studies have identified hospital sinks as reservoirs of Gram-negative bacteria. The aim of this study was to prospectively investigate the bacterial transmission from sinks to patients and if self-disinfecting sinks could reduce this risk. Samples were collected weekly from sinks (self-disinfecting, treated with boiling water, not treated) and patients in the Burn Centre at Linköping University Hospital, Sweden. The antibiotic susceptibility of Gram-negative isolates was tested, and eight randomly chosen patient isolates and their connected sink isolates were subjected to whole genome sequencing (WGS). Of 489 sink samples, 232 (47%) showed growth. The most frequent findings were *Stenotrophomonas maltophilia* (*n* = 130), *Pseudomonas aeruginosa* (*n* = 128), and *Acinetobacter* spp. (*n* = 55). Bacterial growth was observed in 20% of the samplings from the self-disinfecting sinks and in 57% from the sinks treated with boiling water (*p* = 0.0029). WGS recognized one transmission of *Escherichia coli* sampled from an untreated sink to a patient admitted to the same room. In conclusion, the results showed that sinks can serve as reservoirs of Gram-negative bacteria and that self-disinfecting sinks can reduce the transmission risk. Installing self-disinfecting sinks in intensive care units is an important measure in preventing nosocomial infection among critically ill patients.

## 1. Introduction

Healthcare-related infections caused by multidrug-resistant Gram-negative bacteria are medically challenging. Few treatment options are usually available due to the wide and complex range of resistance mechanisms these bacteria carry [1,2], and, as a consequence, they are associated with an increased financial burden, prolonged hospital stays, and increased mortality [3,4].

In intensive care units (ICUs), the clinical impact of opportunistic Gram-negative bacteria with multidrug resistance, such as *Acinetobacter baumanii*, *Enterobacter cloacae*, *Klebsiella pneumoniae*, *Pseudomonas aeruginosa*, and *Stenotrophomonas maltophilia*, is increasing. Medical conditions associated with these bacteria range from the colonization of the respiratory and urinary tract to deep and disseminated infections [5,6,7].

In this context, burn patients represent an especially difficult cohort. A loss of a functioning skin barrier in the form of a third-degree burn, often combined with an inhalation injury and endotracheal intubation, entails a dysfunctional immune system and a high vulnerability to the colonization of Gram-negative bacteria. Infections are frequent and can lead to everything from melting skin grafts to septic shock and death [8]. The more severe or larger the burn injury is, the more likely it is that an infection will ensue. To prevent serious complications, it is essential to have a proactive approach and treat the infection as early and efficiently as possible. Cultures are therefore often regularly performed and repeated courses of antibiotics are prescribed. The high selective pressure favours multidrug resistance, and common bacterial findings are *Acinetobacter* spp., *K. pneumoniae*, and *P. aeruginosa* [9,10,11].

With a few exceptions, Gram-negative bacteria are sensitive to dehydration. They are therefore typically found in moist environments, e.g., sinks and their drain systems. The water traps of sinks constitute a relatively protected environment, which favours the growth of bacteria and production of biofilms [12,13,14]. Once biofilms have been established, disinfectants cannot fully eradicate them [15,16]. Through splash water and aerosols, bacteria can be mobilized and transmitted from the sinks to patients. Sinks have been identified as potential sources of infections and outbreaks in ICUs in several reports, but their clinical importance has, to some extent, been questioned due to the lack of prospective studies available [17].

A burn centre is a complex and stressful care environment [18]. Operations are usually performed in the patient room to avoid moving the patient, and the patient may stay for several months. Thus, the sinks located in the patient rooms are frequently used for purposes other than hand washing. Gram-negative bacteria therefore tend to accumulate in the sinks and their drain systems. To explore the extent of sink contamination, samples from water traps in sinks at the Burn Centre at Linköping University Hospital, Sweden, were cultured during the summer of 2018. The growth of clinically relevant Gram-negative bacteria was recorded in all sinks placed in patient rooms and the associated bathrooms. Furthermore, several of the identified species were also observed in blood and wound samples from admitted patients.

The aim of this study was to investigate if it would be possible to reduce the load of Gram-negative bacteria in sinks, and thereby also indirectly the risk of nosocomial infections in a burn centre, by installing newly developed self-disinfecting sinks. The design of the study also made it possible to prospectively explore the transmission of Gram-negative bacteria from sinks to patients.

## 2. Materials and Methods

### 2.1. Settings

The Burn Centre at Linköping University Hospital, Östergötland County, Sweden, is one of two units for national highly specialised care of severely burned patients in Sweden. Approximately 100 patients are admitted each year. The catch area is nationwide, but the majority of patients are referred from the south of Sweden. The unit offers a total of seven single-bed rooms, of which four (rooms 1–4) are equipped for intensive care with a high level of medical monitoring and access to respiratory care. There are two sinks per room: one located in the patient room and the other in the bathroom. The sinks are used for hand washing, for the cleaning of various medical devices, and in direct patient care.

Since the study material only comprised bacterial isolates and no changes were made in well-established clinical routines, no ethical approval was sought.

### 2.2. Self-Disinfecting Sinks

The self-disinfecting stainless-steel sink (Dissinkfect^®^, Micropharmics AB and Tunerlux AB, Uppsala, Sweden) used in this study has a built-in heating supply, which heats the wash bowl to 75 °C and the water trap to 100 °C (Figure 1). It tolerates quick temperature changes and is commonly used for cleansing or disinfecting agents. By pressing a button placed on the side of the sink for four seconds, the disinfection process starts and a green LED indicator shines during the whole 15 min process. It can be stopped at any time and its length and temperatures can be adjusted according to the requirements. During the study period, self-disinfection was initiated once per each work shift, i.e., three times every 24 h.

Two self-disinfecting sinks, one located in the patient room and one in the bathroom, were installed in room 1. This was the intensive care room most frequently occupied by patients prior to the study. Another intensive care room (room 4) was selected as a comparator, and the sinks in this room were treated with boiling water (3 L each) once a week during the entire study period. The remaining sinks at the centre acted as controls and their water traps were not disinfected at any time. The external surfaces of all sinks, including faucets and bowls, were cleaned daily with alcohol wipes or cloths dampened with isopropanol.

### 2.3. Environmental Cultures

To explore the growth of different bacteria in the water traps of sinks over time, environmental samples were collected with ESwabs (Copan Diagnostics Inc., Murrieta, CA, USA) from all 14 patient-associated sinks in the Burn Centre. The sampling took place at 8 a.m. every time, i.e., approximately 4 h after the last disinfection cycle. The swabs were inserted through the strainer and turned around. The collection of samples started directly after the installation of the self-disinfecting sinks in September 2019 and continued on a weekly basis until April 2020, for a total of 35 weeks. Records were kept concerning patient occupancy of each room upon sampling.

The samples were sent to the Department of Clinical Microbiology, Linköping University Hospital, and streaked onto three different types of media using the swabs: blood agar, hematin agar, and chromogenic urinary tract infection (UTI) agar (Thermo Fisher Scientific, Waltham, MA, USA). Discs (Thermo Fisher Scientific, Waltham, MA, USA) with imipenem (10 µg), trimethoprim–sulfamethoxazole (1.25–23.75 µg), and linezolid (10 µg) were placed on the plates, respectively. The plates were incubated at 35 °C for approximately 48 h. Bacteria were identified to the species level with a MALDI Biotyper 3.0 (Bruker Corporation, Karlsruhe, Germany).

### 2.4. Patients

All patients admitted to the Burn Centre during the study period were cultured once a week and upon any clinical sign of infection, according to the routines of the unit. ESwabs (Copan Diagnostics Inc., Murrieta, CA, USA) were used when sampling from burn wounds. The samples were streaked onto four different types of media using the swabs: hematin agar, chromogenic UTI agar, streptococcus agar, and chromogenic *Staphylococcus aureus* (CSA) agar (Thermo Fisher Scientific, Waltham, MA, USA), and incubated at 35 °C for approximately 48 h. All Gram-negative isolates were frozen at −70 ℃ to allow for future genetic analyses. Gram-positive isolates were only frozen if the quality manual of the laboratory indicated to do so. During the patient’s stay, it was recorded in which room the patient was placed. The relocation of patients was avoided, unless a patient was moved from a room equipped for intensive care to a regular room as a result of treatment progress.

### 2.5. Antibiotic Susceptibility Testing

The antibiotic susceptibility was tested with the disc diffusion method according to the recommendations of EUCAST (www.eucast.org, accessed on 11 October 2022). For environmental Gram-negative bacteria, testing was conducted with cefotaxime, ceftazidime, cefepime, piperacillin–tazobactam, imipenem, meropenem, nalidixic acid, ciprofloxacin, tobramycin, and trimethoprim–sulfamethoxazole. For *P. aeruginosa*, the susceptibility testing was limited to ceftazidime, imipenem, and meropenem, and for *S. maltophilia*, to trimethoprim–sulfamethoxazole.

Patient isolates were tested when judged clinically relevant and against antibiotics recommended for each species. To comply with the Swedish Infection Protection Act, cefoxitin-resistant *S. aureus* isolates were further analysed with a polymerase chain reaction (PCR) to determine the carriage of the *nuc* gene and the *mecA* gene. All methicillin-resistant *S. aureus* (MRSA) isolates were subjects for whole-genome sequencing (WGS).

An isolate was considered multidrug-resistant if it was resistant to at least three classes of antibiotics, although *P. aeruginosa* and *S. maltophilia* were exempted. *Enterobacterales* with reduced susceptibility to cefotaxime/ceftazidime/cefepime were further phenotypically evaluated with gradient diffusion tests containing cefotaxime/ceftazidime/cefepime, with and without clavulanic acid (Thermo Fisher Scientific, Waltham, MA, USA). *Escherichia coli* isolates that were resistant to cefoxitin and exhibited no effect of clavulanic acid were tested with gradient diffusion tests containing cefotetan, with and without cloxacillin (Thermo Fisher Scientific, Waltham, MA, USA). Unclear outcomes were explored with WGS to clarify the resistance mechanism(s).

### 2.6. WGS

Eight isolates were randomly chosen from the same number of patients, and isolates from water traps of sinks that belonged to the same species and were connected in space and time to each patient were subjected to WGS. Furthermore, there were rooms in which a patient was colonized by the same species as the former patient, but there was no growth of this species in the sinks. Five of these patients’ isolates were randomly selected for sequencing, together with five isolates from former patients. The patients had stayed in rooms 1, 2, 4, 5, and 6.

DNA was prepared from 1 µL from a single colony of each isolate, using the EZ1 DNA Tissue Kit (Qiagen, Germantown, MD, USA), with an included pre-heating step at 95 °C and shaking at 350 rpm. Twenty nanograms of DNA was used for library preparation, using the QIAseq FX DNA Library Kit (Qiagen, Germantown, MD, USA) with an 8 min fragmentation time. DNA libraries were sequenced on the MiSeq platform (Illumina, San Diego, CA, USA) with 2 × 300 bp paired-end reads.

Data analysis was performed in CLC Genomics Workbench v. 10.1.1 with the Microbial Genomics Module v. 2.5.1 (Qiagen, Germantown, MD, USA). Multilocus sequence typing (MLST) analysis was performed using the PubMLST (pubmlst.org, accessed on 19 October 2022) scheme for each randomly chosen bacterial species. Read mapping and variant calling were performed against the different reference genomes with NCBI accession numbers NC_008253 (*E. coli*), NC_018405 (*E. cloacae*), NC_017548 (*P. aeruginosa*), and NC_010943 (*S. maltophilia*), with the following thresholds to call a variant: depth of coverage ≥20×, frequency ≥90%, and Phred score ≥20. A quality filter was then applied that retained variants with a sequencing depth of ≥20× in all samples and a distance ≥10 bp to the next variant. The resulting variants were used to create single-nucleotide polymorphism (SNP) trees and calculate the genetic distances between samples. Previous studies suggest that isolates of *E. coli* and *P. aeruginosa* with distances of ≤10 SNPs and ≤37 SNPs, respectively, are likely to belong to the same clone [19]. So far, no studies have suggested SNP thresholds for *E. cloacae* or *S. maltophilia*, but an SNP distance of <21 has been considered to support the notion that two bacterial isolates in general have arisen from the same source [20].

### 2.7. Statistical Analysis

Fischer’s exact test was used when comparing the culture results from the three groups of sinks (self-disinfecting, treated with boiling water, not treated). A *p*-value of ≤0.05 was considered statistically significant.

## 3. Results

### 3.1. Environmental Samples

A total of 489 samples were collected from the water traps of the sinks during the study period. Of these, 232 samples (47%) showed the growth of one or more bacterial species. The three most frequent Gram-negative bacteria were *S. maltophilia* (*n* = 130), *P. aeruginosa* (*n* = 128), and *Acinetobacter* spp. (*n* = 55). For more details, see Figure 2. The growth of Gram-positive bacteria consisted mostly of skin flora: coagulase-negative staphylococci (*n* = 24), *S. aureus* (*n* = 1), and *Enterococcus faecalis* (*n* = 6).

Bacterial growth in one or both of the self-disinfecting sinks located in room 1 was observed on seven (20%) different sampling occasions. The bacterial load in these sinks was significantly lower than in those treated with boiling water once a week (*p* = 0.0029) and those that were not treated at all (*p* = < 0.00001). The total number of Gram-negative isolates was eleven and consisted of *Acinetobacter* spp. (*n* = 5), *S. maltophilia* (*n* = 4), and *P. aeruginosa* (*n* = 2).

In the sinks treated with boiling water in room 4, 57 Gram-negative bacterial isolates belonging to 7 bacterial genera were collected on 20 (57%) different sampling occasions. The sinks located in the remaining rooms (no disinfection treatment) showed the broadest range of bacterial species and an even higher proportion of bacterial growth (Figure 2).

The distribution of bacteria in the water traps of the sinks in the patient rooms and the bathrooms varied. In room 1, the majority (91%) of the bacteria were sampled from the bathroom. The corresponding figures for rooms 2–7 were 46%, 39%, 42%, 51%, 54%, and 70%, respectively.

The occupancy of the rooms in the Burn Centre differed. The room with the highest level of occupancy was room 2. It was occupied by four patients during 29 of the study weeks (83%). In contrast, room 7 was only occupied during two weeks (6%) and by two patients. This was the lowest level of occupancy. The remaining rooms were occupied as follows: room 1 by four patients during 23 weeks (66%), room 3 by seven patients during 18 weeks (51%), room 4 by five patients during 22 weeks (63%), room 5 by eight patients during 17 weeks (49%), and room 6 by six patients during 25 weeks (71%). In all rooms but room 1, an increased accumulation of bacteria was observed when a patient was admitted to the room.

The antibiotic susceptibility testing revealed a multidrug-resistant *E. coli* strain sampled from sinks in room 4. It was ESBL-producing; was resistant to cefotaxime, ceftazidime, cefepime, piperacillin–tazobactam, nalidixic acid, ciprofloxacin, tobramycin, and trimethoprim–sulfamethoxazole; and had been brought into the unit by the patient staying in the room. It was detected in the sinks for four weeks. After the patient was discharged, the two sinks were treated with boiling water and no new patient was admitted until the cultures were negative. *P. aeruginosa* isolates with resistance to ceftazidime, imipenem, and meropenem were observed on different sampling occasions from sinks in rooms 1, 2, and 4. The remaining isolates showed no deviant resistance patterns.

### 3.2. Patient Samples

A total of 36 patients were admitted to the Burn Centre during the study period. The duration of the stay varied depending on the severity of the burn injuries, e.g., room 2 was occupied by the same patient for 20 weeks before relocation, whereas another patient stayed for less than one week in room 5.

Culture samples collected from the patients showed the following growth of Gram-negatives: *P. aeruginosa* (*n* = 31), *E. cloacae* (*n* = 28), *E. coli* (*n* = 11), *Klebsiella* spp. (*n* = 7), *Proteus* spp. (*n* = 6), *S. maltophilia* (*n* = 6), *Acinetobacter* spp. (*n* = 3), *Serratia marcescens* (*n* = 2), *Citrobacter freundii* (*n* = 2), *Morganella morganii* (*n* = 1), *Enterobacter amnigenus* (*n* = 1), and *Moraxella catarrhalis* (*n* = 1). The growth of Gram-positive bacteria consisted of *S. aureus* (*n* = 274), coagulase-negative staphylococci (*n* = 88), *Enterococcus* spp. (*n* = 89), *Streptococcus* spp. (*n* = 37), and *Bacillus* spp. (*n* = 13).

Multidrug-resistant bacteria isolated from patients included seven samples of MRSA isolated from two patients admitted to room 6 on different occasions (unrelated strains which were never found in any sink) and the multidrug-resistant *E. coli* found in the sinks in room 4. It was isolated from the patient at admittance. A *P. aeruginosa* strain with resistance to ceftazidime, ciprofloxacin, imipenem, meropenem, and piperacillin–tazobactam was isolated at several different sampling occasions from a patient admitted to room 2. This patient was also colonized by an *E. cloacae* strain resistant to cefotaxime, ceftazidime, and piperacillin–tazobactam.

### 3.3. WGS Results

A total of 24 isolates were subjected to WGS: *E. cloacae complex* (*n* = 8), *P. aeruginosa* (*n* = 8), *E. coli* (*n* = 4), and *S. maltophilia* (*n* = 4). Two of the *P. aeruginosa* genomes were used twice, i.e., they were not only included when comparing sink–patient genomes but also when comparing patient–patient genomes when there was no growth in the bacterium in the sinks.

The samples obtained an average sequencing depth of 64×. One cluster was recognized with MLST and whole-genome-wide phylogenetic analysis. The cluster contained two isolates of *E. coli*: one sampled from a patient placed in room 6 and the other was an environmental sample collected from one of the sinks in the same room one week earlier. The isolates had a difference of a single SNP and were identified as sequence type (ST) 625. The remaining isolates all belonged to unique clones.

Isolates identified with MALDI-TOF mass spectrometry as *E. cloacae* complex constituted a special problem. In half of the cases, the genomes that were compared did not belong to the same species. Species within the complex identified with WGS included *Enterobacter roggenkampii*, *Enterobacter hormaechei*, and *Enterobacter ludwigii*.

The *E. cloacae* complex is known for its ability to harbour the plasmid-mediated *sil* operon, a gene cluster encoding efflux pumps, a silver-binding protein, and regulatory genes that confer resistance to silver [21]. Silver products are often used in burn centres and could therefore select for this bacterial complex, which was a relatively frequent finding in both sink and patient samples. The genomes of isolates belonging to the *E. cloacae* complex were therefore screened for the *sil* operon [21]. Six out of eight isolates (75%) carried the full operon.

## 4. Discussion

There has been a clear increase in sink-associated outbreaks caused by Gram-negative bacteria in recent years [22,23,24,25,26,27]. In the present study, it was investigated if stainless steel sinks, in which both the bowl and the water trap were self-disinfected three times per 24 h, could reduce the bacterial load and thereby the risk of transmission. Furthermore, two conventional sinks were treated weekly with boiling water as an easy and cheaper alternative. The results showed that both alternatives reduced the bacterial load of the sinks compared to no disinfection at all, but the self-disinfecting sinks were significantly more efficient. This is in accord with other studies in which self-disinfecting sink drains have been used [28,29].

The self-disinfecting sinks in room 1 had the overall lowest frequency of bacterial growth and the lowest number of species isolated during the entire study period. In contrast to all the other rooms, there was no correlation between patient occupancy and the bacterial growth in the sinks; the bacterial load remained low or was zero despite a suboptimal use of sinks during patient care. Although the routine to initiate a self-disinfecting cycle every 8 h did not eliminate all bacterial growth, it showed that bacterial growth could be radically decreased. It is quite possible that the bacterial growth would have been further reduced if the self-disinfecting cycle had been started every time the sink was contaminated. The health care personnel at the centre found, however, that this instruction was too complicated and time-consuming, which is why it was changed to every 8 h.

Treatment with boiling water was a simple and functioning alternative that kept the bacterial load at a relatively low level in the sinks located in room 4. As shown in a study from 2021, the initial concentration of bacteria in the drain is back within approximately a week [30]. Thus, this alternative needs to be carried out at least once weekly and continuously to avoid the re-occurrence of growth. In addition, the procedure involves extra workload for the personnel and there is always a risk of contracting burn injuries while handling the boiling water. It was, however, chosen over chlorine, the traditional disinfectant for hospital sinks, since it has been shown to be 100 to 1000 times more effective in reducing pathogens, does not smell, is environmentally friendly, and is fairly inexpensive [30]. The replacement of contaminated sinks has been shown to reduce the infection rates in ICUs [31,32], but bacteria may not only reside in the water trap. They can also be found further down in the drain system. As a result, bacteria can reappear despite a complete change of sinks [25]. Self-disinfecting sinks are therefore a better and the most long-term solution to the problem.

The whole-genome-wide phylogenetic analysis identified one cluster among the 24 patient and environmental samples that were randomly chosen. The cluster consisted of two *E. coli* isolates belonging to ST625, a clone associated with extra-intestinal infections [33]. The sink isolate was collected one week earlier than the patient isolate, indicating that the sink was the likely source of the bacterium that colonized the patient’s burn wounds. This is, to our knowledge, the first time this type of event has been observed prospectively in a clinical setting. The exact route for the transmission is, however, not clear. Few studies deal with the exact mechanism of transmission from a sink to a patient. In a recent study, the mobilization of bacteria from biofilms in the water traps of sinks to the surrounding environment was demonstrated by using green-fluorescent-expressing *E. coli* [12]. This is a possible transmission route for the *E. coli* in room 6.

Additional transmissions may also have occurred in this and other rooms, but the low number of isolates investigated and the fact that only a single colony was used when preparing the DNA limited the chances of detecting them. Interestingly, in the five cases in which a patient was colonized by the same Gram-negative species as the former patient, and as the sinks lacked growth in the species of interest, no transmission was observed. However, the number of colonies/isolates investigated may once again have been too low.

There were few multidrug-resistant isolates in the present study, but resistance does not always come in the form of antibiotic resistance. The isolation frequency of *E. cloacae* complex was relatively high among the Gram-negatives. Only one isolate was resistant to more broad-spectrum beta-lactams, whereas the carriage rate of the *sil* operon was quite high, at 75%. In an earlier study [21], 48% of invasive *E. cloacae* isolates harboured *sil* genes. These findings suggest that the use of silver products rather than antibiotics could have selected for this complex, but whether or not the genes were expressed was never tested.

Although the main focus of this study was on Gram-negative bacteria, it was striking how few Gram-positive bacteria were isolated from the sinks compared to from the patients. For instance, only a single *S. aureus* isolate was recorded from the sinks. The corresponding figure from patients was 274, indicating that water traps mainly offer an environment that promotes the growth of Gram-negative bacteria, and of *S. maltophilia* and *P. aeruginosa* in particular. However, even if *S. aureus* did not thrive in the water traps, it may survive, together with other Gram-positive bacteria and *Acinetobacter* spp., in the wash bowl. To reduce the risk of dissemination from this part of the sink, the wash bowl was also decontaminated during the disinfection process.

In conclusion, the results prospectively showed that sinks can serve as a reservoir for Gram-negative bacteria, and that self-disinfecting sinks can reduce the bacterial load in the sinks and thereby also the risk of bacterial transmission. Installing self-disinfecting sinks in ICUs is therefore an important measure in preventing nosocomial infection among critically ill and vulnerable patients. A less expensive but less efficient solution can be to disinfect sinks with boiling water once weekly.

## Figures and Tables

**Figure 1 microorganisms-11-00965-f001:**
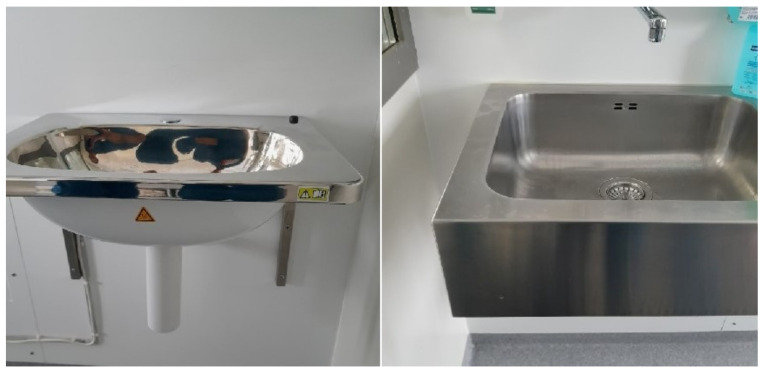
The self-disinfecting sink installed in room 1 is shown to the left, whereas the regular sink used in all of the other rooms in the Burn Centre is shown to the right.

**Figure 2 microorganisms-11-00965-f002:**
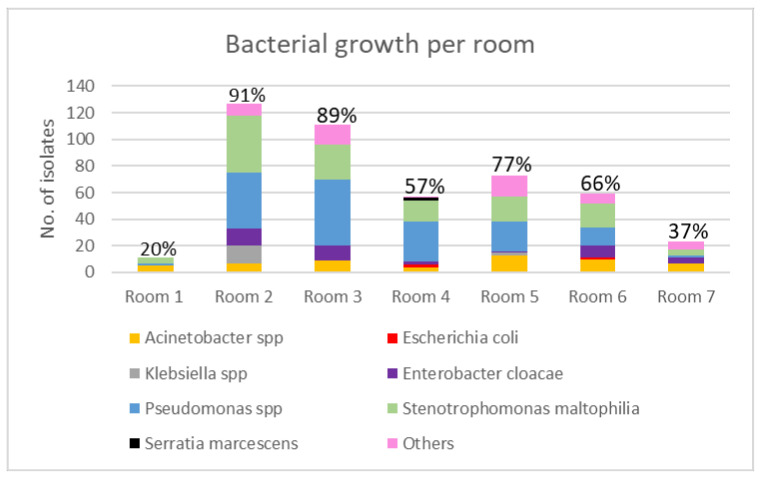
The number of bacterial isolates and types of bacteria are shown per room. The percentages of sampling weeks with bacterial growth are shown above the bar for each room. All rooms had two sinks. The self-disinfecting sinks were installed in room 1, the sinks in room 4 were treated weekly with boiling water, and the sinks in the remaining five rooms were untreated.

## Data Availability

More detailed patient data are not openly available to ensure the confidentiality, but are available from the corresponding author upon reasonable request.

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
