# Peer review of "Source Control of Gram-Negative Bacteria Using Self-Disinfecting Sinks in a Swedish Burn Centre"

_microorganisms, 2023, doi:10.3390/microorganisms11040965_

Round 1
Reviewer 1 Report
This manuscript describes three independent but interconnected pieces of work:
1-Longitudinal weekly sampling of 14 patient sinks located in 7 patient rooms in a Burn Centre over 35 weeks between September of 2019 and April 2020.
2-Comparison of sink-patient genomes when transmission from sink to patient might have occurred and patient-patient genomes for patients that were colonized by the same species as a previous patient after staying in the same room.
3-Evaluation of engineering/disinfection strategies for the sinks in the unit, including the installation of two self-disinfecting sinks in one patient room and the application of boiling water weekly to another patient room.
I commend the authors on completing an extensive prospective sampling campaign (489 sink samples) and subsequent investigation of possible transmission of pathogens from sinks to patients using whole genome sequencing (pieces of work 1 and 2). Some clarification is needed about the methodology used and the presentation of results, as specified in the comments below. However, I believe that the study design is not appropriate to determine whether the self-disinfecting sinks provide a benefit in this setting (piece of work 3). The first issue is that the sampling started after the sinks in room 1 were exchanged by the self-disinfecting sinks. Therefore the levels of contamination of the sink before the intervention are unknown. As the authors themselves acknowledge (line 300), replacement of sinks has been associated with reduction of drain contamination and infection rates in the literature (another e.g. The sink as a potential source of transmission of carbapenemase-producing Enterobacteriaceae in the intensive care unit - PMC (nih.gov)). In this study, there were no appropriate controls to determine whether the effect seen was a result of the action of the self-disinfecting sink or just the replacement of the sink per se, as only the sinks of patient room 1 were replaced by the self-disinfecting sinks. In other words, it is unknown whether a similar effect would have been seen if the sinks in room 1 had been replaced by a brand new sink of the same model as those in use in the hospital, especially as there is little detail provided of the design of either sink (e.g. position of tap in relation to drain etc.). Without appropriate controls (monitoring of the sinks that were replaced beforehand and/or replacing other sinks with a different sink model) it might be misleading to state that “self-disinfecting sinks can reduce the transmission risk” (line 22). I suggest a major revision of the manuscript that focuses on the results from the longitudinal sampling and the comparison of isolates (pieces of work 1 and 2) and describes the observational data around the installation of the self-disinfecting sinks acknowledging the limitations of the study design and the uncertainty around the interpretation of the data.
With that in mind, please consider the following points in the revision:
Line 11- Modify abstract to remove the focus from the self-disinfecting sinks
Line 55- Consider changing reference 15 to other paper(s) that have specifically evaluated disinfectants in the drainage system, such as:
https://www.cambridge.org/core/journals/infection-control-and-hospital-epidemiology/article/effectiveness-of-a-hydrogen-peroxide-foam-against-bleach-for-the-disinfection-of-sink-drains/EA61EAD76EB8945D94D89C1688B73840
https://www.cambridge.org/core/journals/infection-control-and-hospital-epidemiology/article/effectiveness-of-foam-disinfectants-in-reducing-sinkdrain-gramnegative-bacterial-colonization/AEC0BD1290A9F9391114818A1F8FCC74
https://www.ncbi.nlm.nih.gov/pmc/articles/PMC5314675/
Line 55- There have been a couple of papers that have determined that that dispersal from sinks to patients occurs through “splashing” rather than aerosols.
https://www.ncbi.nlm.nih.gov/pmc/articles/PMC6504032/
https://pubmed.ncbi.nlm.nih.gov/30367005/
Lines 70-74 Reword to remove the focus from the self-disinfecting sink.
Line 88 Could you provide a picture of the self-disinfecting sink and a picture of the other sink model used in the hospital?
Line 101 Could you provide more details about the daily cleaning of sinks? What type of product was used (e.g. chlorine-based disinfectant, detergent)? Did the cleaning include any attempt of drain decontamination or were only external surfaces (faucets, basin) cleaned?
Line 104 Could you clarify which area of the sink was sampled? I struggle to see how you can sample the water trap with an ESwab, perhaps because there are no pictures of the sinks.
Line 110-115 Could you clarify the rationale for the media and antibiotic discs chosen? Why were antibiotic discs added to the media? How much volume of liquid amies was inoculated on each plate? Which colonies were chosen for identification? All colonies? All different morphologies? A random selection?
Line 118 Rephrase- do you mean to say that burn wounds from all patients were sampled and cultured once a week?
Line 137 For those people that do not have a clinical background, could you clarify the testing rationale for MRSA? Alternatively just simplify as MRSA does not bear a lot of importance in the results section.
Line 144 Could you clarify what tests were used to confirm ESBL production for the E. coli mentioned in the results?
Line 144 change “fenotypically” for “phenotypically”
Line 147 Specify what isolates were chosen.
Line 176 As explained above, I don’t believe the not treated sinks in this study are appropriate controls for the intervention of installing self-disinfecting sinks. I suggest the statistical analysis is removed from the manuscript.
Figure 1 There are a number of issues with this figure:
-The shades of blue chosen are too similar to each other and therefore indistinguishable (e.g. it’s not possible to tell apart Klebsiella spp. from Enterobacter cloacae).
-The figure reports on the number of isolates recovered, but we don’t know whether all colonies seen on plates were identified.
-It would be useful to indicate in the figure legend what room had the self-disinfecting sinks and what room had the treatment with boiling water, along with the percentage of occupancy for each room.
-“The percentage of bacterial growth in correlation with the number of sampling occasions” can be simplified to “The percentage of sampling weeks with bacterial growth”
Figure 2 I don’t understand how this percentage was calculated. You should indicate how the calculation has been done in the methods and/or figure legend.
Line 223 Could you be more specific? How long was the ESBL-producing E. coli detected for? Was the treatment with boiling water effective?
Line 255 This is a really interesting result and there are very few examples in the literature where the study design has enabled to look into sink to patient transmission. When the E. coli was isolated from the sink, was the patient already in that room or was the room empty/another patient admitted? How confident are you that the patient did not already have the E. coli but it might have gone undetected?
Line 260 Very good point about the difficulties of investigating the relation between isolates belonging to the E. cloacae complex based on MALDI-TOF identification.
Line 268 No mention in the methods about how you screened for the sil operon.
Discussion
Line 296 Is there any risk of burning injuries when using the self-disinfecting sink with the basin heating up to 75 C?
Line 303 A strength of this study is the long sampling period. Do highlight that number of isolates recovered remained low in the self-disinfecting sink but acknowledge that no appropriate controls were used and therefore the impact of simply replacing the sink is unknown.
Line 310 You might want to check out the following paper where the authors make a similar observation.
Genomic surveillance of Escherichia coli and Klebsiella spp. in hospital sink drains and patients - PubMed (nih.gov)
Line 312 There is also a paper where the authors investigated how the position of the drain in relation to the tap and the drainage conditions impact dispersal from sinks.
Carbapenem-resistant Enterobacteriaceae dispersal from sinks is linked to drain position and drainage rates in a laboratory model system - PMC (nih.gov)
Reviewer 2 Report
This paper provides valuable data on practical measures to reduce bacterial contamination in hospital sinks, namely self-disinfecting sinks, and thus reduce the potential for wound infection in highly susceptible patients. This evidence-based approach is important in justifying the expenditure for installing such devices. The use of whole genome sequencing demonstrated association between environmental and clinical isolates and antibiotic resistance testing further underlined the importance of reducing patients' potential exposure. The paper also demonstrated that an alternative intervention, boiling water treatment, also reduced contamination although not to the same extent as self disinfecting devices.
I have just a few minor comments as follows
Page 2 line 51 'subject to desiccation'. Semantics, but desiccation is an active, intentional process. This should be 'dehydration' the inadvertent act.
Page 2 line 85 'no ethical approval'. I'm sure you were advised correctly, but I'm surprised it was not needed as the interventions meant that patients not having the sink disinfection regime could be placed at greater risk of infection. I accept that the baseline 'no intervention' meant that patients' conditions were not made worse by the study, but some patients were denied something that made it better.
Page 3 line 103 - were swabs used just to take water from the traps or were they directed toward the internal surfaces of the traps, thus collecting biofilm? This may make a difference as biofilm is likely to yield a richer microbiome. Presume sampling was done by putting the swab down the plug hole, not by for example a sampling port in the sink trap. Please clarify how samples taken. It says (line 100) that "The remaining sinks at the centre acted as controls and were not disinfected at any time. All sinks were cleaned daily". Cleaning of sinks will result in some disinfectant product going into the sink trap which could have at least a temporary effect on trap microbiome. Was any attempt made in the sampling protocol to separate sampling times from sink cleaning times?
Figure 2 on page 6 - I think this needs more explanation as to what we are seeing to ensure it adds value to the paper.
Page 8 line 304 'and solution' - I think the 'and' needs to be removed.
